

# Evidence of overfishing of geoduck clam *Panopea globosa* from a length-based stock assessment approach

Marlene A. Luquin-Covarrubias[1], Enrique Morales-Bojórquez[1], Juan A. García-Borbón[2], Sergio Amezcua-Castro[3], Sergio A. Pérez-Valencia[4,5] and Estefani Larios-Castro[1]

[1] Centro de Investigaciones Biológicas del Noroeste, La Paz, B.C.S., México
[2] Instituto Nacional de Pesca y Acuacultura (INAPESCA), CRIAP, La Paz, B.C.S., Mexico
[3] Centro Interdisciplinario de Ciencias Marinas, La Paz, B.C.S., México
[4] Centro Intercultural de Estudios de Desiertos y Océanos A.C., Puerto Peñasco, Sonora, México
[5] Comisión Nacional de Áreas Naturales Protegidas, Hermosillo, Sonora, México

## ABSTRACT

Stock assessment of the geoduck clam *Panopea globosa* in Mexico has been based on data-poor without consideration of the biological traits of the species, promoting a passive management strategy without biological reference points for its harvest and conservation, which results in limited advice regarding the sustainability of the fishery. The stock assessment was supported on an integrated catch-at-size assessment model. The model described the population changes, including recruitment, selectivity, fishing mortality, individual growth patterns and survival over time, providing management quantities for the geoduck clam fishery, such as biomass-at-length (total and vulnerable) and harvest rate-at-length. The results indicated overfishing of the geoduck clam population; the harvest rate exceeded the management tactics established for this fishery, even the individuals smaller than the minimum legal size (130 mm) were harvested. Thus, declines in the total biomass (from 3,262 to 1,130 t) and recruitment (representing an 86% decrease) were observed from 2010 to 2012. Although the results showed a recovery trend in recruitment and total biomass from 2014 to 2016, this trend may have been due to the spatial relocation of fishing mortality.

## INTRODUCTION

Stock assessment and fishing management are crucial when starting a new fishery because there is high uncertainty whether the catch levels will be sustainable; thus, increases in the fleet size and fishing effort could lead to depletion of the stock, which occurs when commercial catches decrease under an optimal economic yield (*Branch et al., 2006*). Moreover, delayed implementation of management reforms has negative effects on stock conditions and fishery benefits; it significantly reduces biomass levels, harvest and profits; in addition, delayed influences on continuous declines, increase the risk of collapse and delay the recovery of stocks (*Mangin et al., 2018*). Collapses have been documented for several stocks of sedentary organisms, such as abalone (*Haliotis* spp.), which are distributed from

Corresponding author
Enrique Morales-Bojórquez,
embojorq@prodigy.net.mx

southeast Alaska to southeast California, where regional management strategies have failed to consider the biology and distribution of the populations (*Hilborn, Orensanz & Parma, 2005*). The zigzag scallop (*Euvola ziczac*) in Brazil was not considered a target species; therefore, management strategies were not developed, and the stock irreversibly collapsed (*Pezzuto & Borzone, 2004*). Finally, the smooth clam (*Callista chione*) in the northwestern Mediterranean Sea declined due to the association between the increase in fishing effort and sand dredging (*Baeta, Ramón & Galimany, 2014*). Similar events occurred for *P. generosa* in Canada and *P. zelandica* in New Zealand, where the total allowable catch was exceeded; thus, moratorium and fishery closures were implemented to stabilize the populations (*Khan, 2006*; *Gribben & Heasman, 2015*).

For long-lived organisms such as the species of the genus *Panopea,* recovery of the stock can be extremely slow. The main disadvantage is that these species have complex population dynamics characterized by prolonged longevity, low recruitment, variable individual growth rates, and high natural mortality rates in young individuals (*Goodwin & Pease, 1991*). Additionally, geoduck populations are composed of many annual classes, suggesting an apparent stable abundance, which makes them particularly vulnerable to depletion (*Orensanz et al., 2004*).

In Mexico, the geoduck clam constitutes a small-scale fishery that has been developed for two species: *Panopea generosa* in the northwest Mexican Pacific and *Panopea globosa* in the Gulf of California and southwest of the Baja California Peninsula. For both species, harvesting increased quickly, reaching landings from 38 to 2,000 t during 2002–2011 (*Aragón-Noriega et al., 2012*). However, one of the main problems for managing this fishery is the lack of key and continuous information that is needed to apply models and reliable stock assessments (*Arreguín-Sánchez & Arcos-Huitrón, 2011*; *Ramírez-Rodríguez & Ojeda-Ruíz, 2012*). According to *Dowling et al. (2015)*, the geoduck clam fishery could be classified into two categories. (1) A data-poor fishery: it has limitations in terms of the type and quality of available data; therefore, quantitative stock assessments are impossible to perform; given that the best available information is inadequate, reference points and the exploitation status of the stock cannot be determined. (2) A data-limited fishery: additional information sources on catch and effort data can be included, but the amount of information is not sufficient to perform a quantitative stock assessment. In addition, in species with complex or unknown life histories, the stock status is unable to be determined.

For *Panopea* spp. in Mexico, the stock assessment has been based on fishery-independent surveys, estimating densities and biomasses through extrapolation for known areas, which have important implications for fishery management because this assessment defines the stock size susceptible to fishing (*Zhang & Hand, 2006*; *Cortez-Lucero et al., 2014*). This stock assessment method was adopted from management schemes applied to geoducks in Washington State, USA, and British Columbia, Canada, in the 1970s (*Goodwin & Pease, 1991*; *Zhang & Hand, 2006*), to establish an initial strategy for regulating the geoduck fishery in Mexico. Thus, during the discovery and growth phases of the geoduck clam fishery, the main dynamic was the entry of new participants, the expansion of fishing capacity, and the exploration of new patches and beds. These activities were regulated through management tactics, such as (1) a minimum legal size of 130 mm shell length; (2)

a maximum allowable catch of 0.5% of the estimated total biomass (predevelopment phase) or 1% from a growth phase of the geoduck clam fishery; and (3) requiring the density of the bed to be exploited to be greater than 0.04 geoducks/m$^2$ (*Aragón-Noriega et al., 2012*). However, these guidelines did not consider the biological traits and population dynamics of geoduck and were limited to determining changes in stock size (e.g., biomass, recruitment), restricting estimates of harvest rate and fishing mortality and failing to provide biological reference points for indicating levels of caution in the fishery; this approach is characterized as passive management scheme (*Cadrin & Pastoors, 2008*; *Fitzgerald, Wilson & Lenihan, 2018*). Eventually, during the development of a fishery, the management strategies must change and adapt to the obtained biological knowledge to improve the assumptions about the status of the stock (*Hilborn & Walters, 1992*). However, stock assessment and fishing management for the geoduck clam fishery have not been updated over the last 16 years of harvest, and the basic assumptions about the stock remain unchanged. Thus, the aim of this study is to analyze the effects of fishing pressure on the population of the geoduck clam *Panopea globosa* in Puerto Peñasco, Sonora, Mexico.

## MATERIAL AND METHODS

### Shell length structure and catch data

Biological data were obtained from Puerto Peñasco, Sonora, in the upper Gulf of California, Mexico (Fig. 1), during two time periods: 2010 to 2012 and 2014 to 2016. A total of 19,445 individuals of *P. globosa* were collected from two sources of information. The first source was a sampling design based on the estimation of the fishing ground. To this goal, a survey was conducted to identify the patches and beds with high abundance; the fishing ground was estimated based on the boundaries of sampling stations, and the area was expressed as km$^2$.

The second source of biological data was obtained from commercial landings, which were endorsed by fishing licenses that establish basic conditions for the extraction activities. During the predevelopment phase of geoduck fishery, a legal requirement for the fishers was annual data collection and analysis of the harvest population; however, when the analysis of the fishing ground indicates the availability of sufficient biomass for exploitation, then the fishing ground is classified as being in growth phase. Consequently, the data collection is not mandatory for the fishers, and the biological information may be limited due to the cost of the geoduck monitoring program, which was not implemented during 2013. According to the bathymetry around Puerto Peñasco (*Ramírez-Mendoza & Álvarez, 2009*), the commercial harvest was performed between 10 and 30 m deep; greater depths are restricted by the Mexican government to avoid the decompression sickness of divers. Geoduck harvest was performed by hookah diving during low tides, with small boats equipped with an outboard motor and a low-pressure compressor connected to a hose. Geoducks were located through their siphon holes on the substrate, and they were removed using a water jet to loosen the sediment. Shell length (mm) and total weight (g) data were recorded monthly; the specimens were grouped by shell length frequency distributions by year, and the size classes varied from 78 to 200 mm shell length.
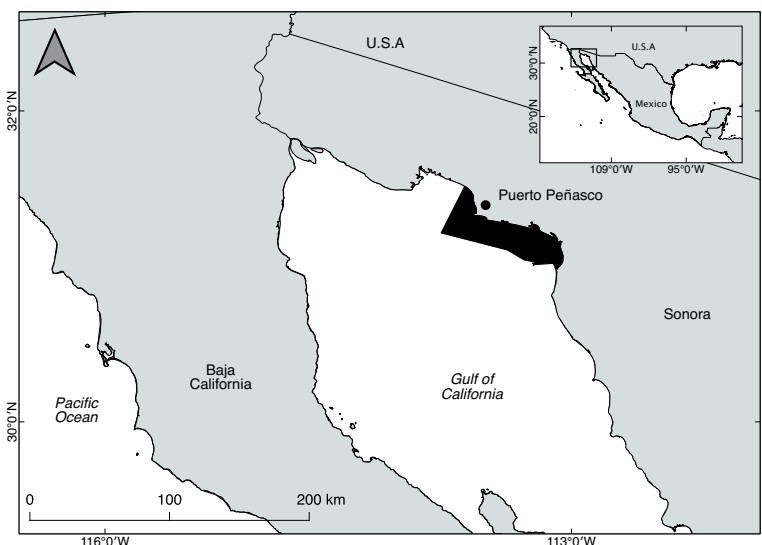

**Figure 1** **Study area for the geoduck clam *Panopea globosa* off Puerto Peñasco, Sonora, Mexico.** The figure was created and edited by Daniela Maldonado Enriquez (CIBNOR SC).

## Data origin

The annual catch data were obtained from the Subdelegación Federal de Pesca at Puerto Peñasco, Sonora, Mexico (Comisión Nacional de Acuacultura y Pesca, CONAPESCA) (https://www.conapesca.gob.mx/wb/cona/estadisticas_de_produccion_pesquera).

Scientific collecting and the use of biological and fishery data was endorsed by the Mexican federal government (CONAPESCA) through the commercial fishery permit numbers DGOPA.05338.050710.3326 (2010), DGOPA.09450.221111.3442 (2011), 126054025018 (2012), PPF/DGOPA-186/14 (2014), 126070025019 (2015), and 126047025017 (2016) with fishermen of the SCPP Buzos de Puerto Peñasco, SCPAP Islas de Sonora, SCPP Jaiberos y Escameros, and SCPP Mar y Tierra del Golfo de Cortez.

## Population dynamics

The population dynamics of *P. globosa* was analyzed using an integrated catch-at-size assessment (ICSA) model (*Morales-Bojórquez, López-Martínez & Beléndez-Moreno, 2013*). To provide a better description of the model, a summary of the symbols and descriptions of the parameters and variables are defined in Table 1. The ICSA model describes the population changes over time for multiple shell length classes according to the following: (a) the total mortality of geoducks in shell length class $l$ at time $t$, representing the sum of fishing and natural mortality ($Z_{l,t} = F_{l,t} + M_{l,t}$), and (b) the relationship between $N_{l,t}$ and $N_{l,t'}$, which denotes the number of geoducks at shell length surviving and growing during the next time period according to the equation $N_{l,t'} = N_{l,t}e^{-Z_{l,t}}$. A natural mortality value of 0.046 was used for modeling the population dynamics of geoduck clam (*González-Peláez et al., 2015a*); this fixed parameter was based on the number of gnomonic intervals determined by dividing the life history of the geoduck into a predetermined number of biological units (life stages). The advantage of this method is that uses the biological

information of the lifespan for knowledge of the ontogenic changes in natural mortality; this procedure is better than estimations based on meta-analysis because these approaches were developed for specific taxa, such as fishes (*Pauly, 1980*), empirical approaches based on biological parameters (*Hoenig, 1983*) or based on life strategy (r-K selection) (*Gunderson, 1980*). A gnomonic interval is a systematic strategy for the unequal subdivision of the lifespan of an individual into time intervals, which increase in duration and proportion for each time interval. Therefore, in terms of elapsed time, two gnomonic intervals in a subdivided life history can be considered equivalent if they each form the same constant proportion of the time elapsed (*Caddy, 1996*; *Ramírez-Rodríguez & Arreguín-Sanchez, 2003*; *Martínez-Aguilar, Arreguín-Sánchez & Morales-Bojórquez, 2005*; *Aranceta-Garza et al., 2016*; *Romero-Gallardo et al., 2018*). Thus, the natural mortality of *P. globosa* was modeled as follows: (1) egg to trochophore larvae (24 h); (2) early larvae (6.5 days); (3) late larvae (11 days); (4) early juveniles (35 days); (5) juveniles (3–9 months); (6) late juveniles (1–2 years); and (7) preadult to adults (3–47 years). Basically, the fishery maintains high fishing pressure on biological unit 7; such that $M = 0.046$ is a plausible value; additionally, the comparison between the gnomonic method and alternative procedures yielded similar values for geoducks larger than 130 mm; therefore, the natural mortality was relatively stable for individuals from preadult to adult (*Calderon-Aguilera et al., 2010*; *González-Peláez et al., 2015a*). The catch $\bar{C}_{l,t}$ was calculated using the Baranov catch equation, $\bar{C}_{l,t} = N_{l,t}\mu_{l,t}$, where $\mu_{l,t}$ is the harvest rate defined as the proportion of individuals that die from fishing mortality compared to total mortality $\mu_{l,t} = F_{l,t}/Z_{l,t}\left(1 - e^{-Z_{l,t}}\right)$.

## Initial conditions

The number of individuals of the geoduck clam population at the beginning of the time period denotes the initial stock abundance, which was estimated as $N_0 = \sum_l N_{l,0}$, where $N_{l,0}$ was estimated as $N_{l,0} = \frac{C_{l,0}(Z_{l,0})}{F_{l,0}(1 - e^{-Z_{l,0}})}$. In this way, the proportion of geoducks in shell length class $l$ was calculated as $P_l = \frac{C_{l,0}Z_{l,0}/F_{l,0}(1-e^{-Z_{l,0}})}{\sum C_{l,0}Z_{l,0}/F_{l,0}(1-e^{-Z_{l,0}})}$. Thus, the shell length distribution of abundance was computed as $\hat{N}_{l,0} = \hat{P}_l N_0$. This procedure simplifies the parameter space for estimating only one $N_0$ instead of estimating a vector of $N_{l,0}$, increasing the performance of the objective function.

## Recruitment

Usually, estimates of recruitment are based on a stock-recruits relationship, as in the studies by *Beverton & Holt (1957)*, *Ricker (1954)* and *Ricker (1975)*. However, establishing a functional stock-recruits relationship for the geoduck clam is very difficult; therefore, recruitment in the *Panopea* genus is only measured in terms of recruitment rates (*Zhang & Hand, 2006*). *Sullivan, Lai & Gallucci (1990)*, *Fisch et al. (2019)* and *Amezcua-Castro et al. (2019)* used length-structured models and proposed that recruitment can be estimated from a gamma probabilistic density function, assuming that it can represent the variability in recruitment. Biologically, recruitment represents the number of individuals at specific age or length classes that are added to the exploitable stock in the fishery (*Myers, 2002*). Recruitment is modeled as the product of a time-dependent variable ($R_t$) corresponding to the annual recruitment and a length-dependent variable ($\varphi_l$), representing the proportion
**Table 1  Symbols and descriptions of parameters and variables used in the ICSA model.**

| Symbol | Description |
|---|---|
| $l$ | Shell length class |
| $t$ | Time (years) |
| $l'$ | Range of shell length classes |
| $t'$ | Initial time period |
| $l_t$ | Shell length of geoduck clam at time $t$ |
| $l_{t+1}$ | Shell length of geoduck clam at time $t+1$ |
| $Z_{l,t}$ | Total mortality at shell length $l$ and time $t$ |
| $F_{l,t}$ | Separable fishing mortality at shell length $l$ and time $t$ |
| $M_{l,t}$ | Constant natural mortality at shell length $l$ and time $t$ |
| $N_{l,t}$ | Abundance of geoducks at shell length $l$ and time $t$ |
| $N_{l,t'}$ | Abundance of geoducks at shell length $l$ and later time $t'$ |
| $N_{l',t'}$ | Total abundance of geoduck at length $l'$ and $t'$ |
| $C_{l,t}$ | Observed catch-at-shell length $l$ and time $t$ |
| $\bar{C}_{l,t}$ | Estimated catch-at-shell length $l$ and time $t$ |
| $\mu_{l,t}$ | Harvest rate at shell length $l$ and time $t$ |
| $N_0$ | Initial population abundance |
| $N_{l,0}$ | Initial abundance of geoducks at shell length $l$ and $t=0$ |
| $\hat{N}_{l,0}$ | Initial estimated abundance of geoduck at shell length $l$ and $t=0$ |
| $P_l$ | Proportion of geoducks at shell length $l$ |
| $s_l$ | Estimated relative selectivity at shell length $l$ |
| $f_t$ | Full-recruitment fishing mortality rate at time $t$ |
| $\alpha_s$ | Selectivity parameter of gamma distribution density function |
| $\beta_s$ | Selectivity parameter of gamma distribution density function |
| $s_{o,l}$ | Observed relative selectivity at shell length $l$ |
| $G_{l,l+1}$ | Growth matrix at shell length $l$ and $l+1$ |
| $S_{l,t}$ | Survival matrix at shell length $l$ and time $t$ |
| $T_{l,t}$ | Transition matrix at shell length $l$ and time $t$ |
| $\Delta_l$ | Growth increments at shell length $l$ |
| $\bar{\Delta}_l$ | Mean growth increment at shell length $l$ |
| $L_\infty$ | Asymptotic shell length |
| $l_*$ | Midlength at shell length $l$ |
| $k$ | Growth rate |
| $\alpha_l$ | Growth parameter of gamma distribution density function at shell length $l$ |
| $\beta_g$ | Growth parameter of gamma distribution density function |
| $R_{l,t}$ | Recruitment to the fishery at shell length $l$ and time $t$ |
| $R_t$ | Time-dependent variable of recruitment |
| $\varphi_l$ | Shell length-dependent variable of recruitment |
| $\alpha_r$ | Recruitment parameter of gamma distribution density function |
| $\beta_r$ | Recruitment parameter of gamma distribution density function |
| $TB_l$ | Total biomass-at-shell length |
| $VB_l$ | Vulnerable biomass-at-shell length |
| $\omega_l$ | Expected weight of geoduck at shell length $l$ |
| $\alpha$ | Parameter of weight-length relationship |
| $\beta$ | Parameter of weight-length relationship |

of the annual recruitment into each length class $l$. This proportion $\varphi_l$ of recruits in each length class was estimated following a gamma distribution with recruitment parameters $\alpha_r$ and $\beta_r$. Thus, the equation for estimating recruitment-at-shell length was $R_{l,} = R_t \varphi_l$ (*Fisch et al., 2019*); this procedure of separating variables allows $R_t$ to be compared with recruitment estimates from standard procedures. According to *Sullivan, Lai & Gallucci (1990)*, recruitment specified in this way represents the type of recruitment observed in nature, where variation in growth, behavior, or food supply can result in individuals entering the population at several length classes. Given that the recruitment to the fishery occurs over a range of shell length classes, we grouped the number of individuals with shell length between 78 and 130 mm (which were caught from the fishery) to represent the recruits added to fishable stock.

## Fishing mortality and selectivity

The assumption in the ICSA model is that the fishing mortality vector can be partitioned into two components: (i) a shell length-specific component that does not vary over time (e.g., a constant exploitation pattern) and (ii) an annual multiplier commonly specified as catchability, vulnerability or selectivity (*Megrey, 1989*). Thus, in this study, the fishing mortality was estimated as a separable product of the shell length selectivity and the full-recruitment fishing mortality rate: $F_l = s_l f_t$. The separability assumption allows the estimation of only a fishing mortality value, which is distributed proportionally in the shell length classes, improving the performance of the model parameterization (*Fisch et al., 2019*). To analyze $s_l$, a gamma probabilistic function was applied; the advantage of this approach is that it provides great flexibility to the model, allowing different selectivity patterns (*Carlson & Cortés, 2003*). Thus, $s_l$ was estimated as $s_l = (l/\alpha_s \beta_s)^{\alpha_s} exp(\alpha_s - l/\beta_s)$. The parameterization was based on the residual sum of squares ($RSS_s$) function:

$$RSS_s = \sum_{i=1}^{n} (s_l - s_{o,l})^2 \tag{1}$$

where $s_{o,l}$ is defined in Table 1.

## Individual growth and survival

Individual growth and survival were modeled by using a transition matrix, which represents the combination of a growth matrix and a survival matrix (Table 1). According to *Cao, Chen & Richards (2016)*, the growth increments assume shell length variability, which can be described as $\Delta_l = l_{t+1} - l_t$. Therefore, the mean growth increments were expressed by the Fabens model (1965): $\bar{\Delta}_l = (L_\infty - l_*)(1 - e^{-k})$. The growth matrix was based on a gamma distribution expressed as $g(\Delta_l | \alpha_l \beta_g) = \frac{1}{\beta_g^{\alpha_l} \Gamma(\alpha_l)} \Delta_l^{\alpha_l - 1} e^{-\Delta_l/\beta_g}$. The expected proportion of individuals growing from shell length class $l$ to shell length class $l+1$ was found by integration of the shell length ranges, $l+1_1, l+1_2$, representing the lower and upper ends of shell length classes, respectively:

$$G_{l,l+1} = \int_{l+1_1}^{l+1_2} g(x | \alpha_l, \beta_g) dx. \tag{2}$$

The survival matrix was estimated as $e^{-Z_{l,t}}$, and the transition matrix was calculated as follows:

$$T_{l,t} = \sum_l G_{l,l+1} S_{l,t}. \tag{3}$$

## Parameter estimation

The ICSA model was fitted through the residual sum of squares (*RSS*) criterion:

$$RSS = \sum_{l,t} (\bar{C}_{l,t} - C_{l,t})^2. \tag{4}$$

The parameters were estimated by minimizing the objective function (*RSS*) with a nonlinear fit using the Generalized Reduced Gradient (GRG) algorithm (*Neter et al., 1996*). The optimization was performed in phases; this procedure consists of estimating only a subset of parameters through the optimization of the objective function and adding more parameters in each sequential phase until all parameters are estimated. An advantage is that to the extent that new parameters are included into a new phase, the previously estimated parameters are still estimated, and they are not fixed as prespecified values, allowing the progressive improvement of the goodness of fit of the *RSS* (*Legault & Restrepo, 1998*; *Luquin-Covarrubias et al., 2016a*; *Luquin-Covarrubias et al., 2016b*). A non-linear fit in phases is an statistical procedure accepted when there is high uncertainty in seed values and where the number of parameters to be estimated is greater than 20 (*Fournier et al., 2012*; *Punt, Huang & Maunder, 2013*; *Canales, Company & Arana, 2016*; *Cao, Chen & Richards, 2016*; *Fisch et al., 2019*). According to *Fournier et al. (2012)*, this procedure is also called "model sculpting". Thus, the $F_{l,t}$ vector was known when $f_t$ was included in the numerical optimization process in the first group of parameters; the second group of parameters included $N_{l,0}$, the third group integrated $R_t$, $\alpha_r$ and $\beta_r$, and the fourth group incorporated $\beta_g$. The $\alpha_s$ and $\beta_s$ parameters were independently estimated and later integrated into the *RSS* as a fifth group. For ICSA model if any parameters are known, then they can be prespecified instead of estimated (*Sullivan, Lai & Gallucci, 1990*); thus, the parameter $L_\infty = 200$ mm was assigned according to the largest observed shell length, $k = 0.25$ was assigned from the von Bertalanffy growth model, reported by *Aragón-Noriega, Calderon-Aguilera & Pérez-Valencia (2015)*, and $M = 0.046$ was assigned based on the report by *González-Peláez et al. (2015a)*.

## Management quantities

In this study, the total abundance-at-shell length of the geoduck clam population was estimated using $T_{l,t}$ and $\hat{N}_{l,0}$. The recruitment-at-shell length was added to represent the population dynamics of the geoduck clam as follows:

$$N_{l',t'} = \sum_l T_{l,t} \hat{N}_{l,0} + R_{l,t} \tag{5}$$

The total and vulnerable biomass-at-shell length were estimated as follows:

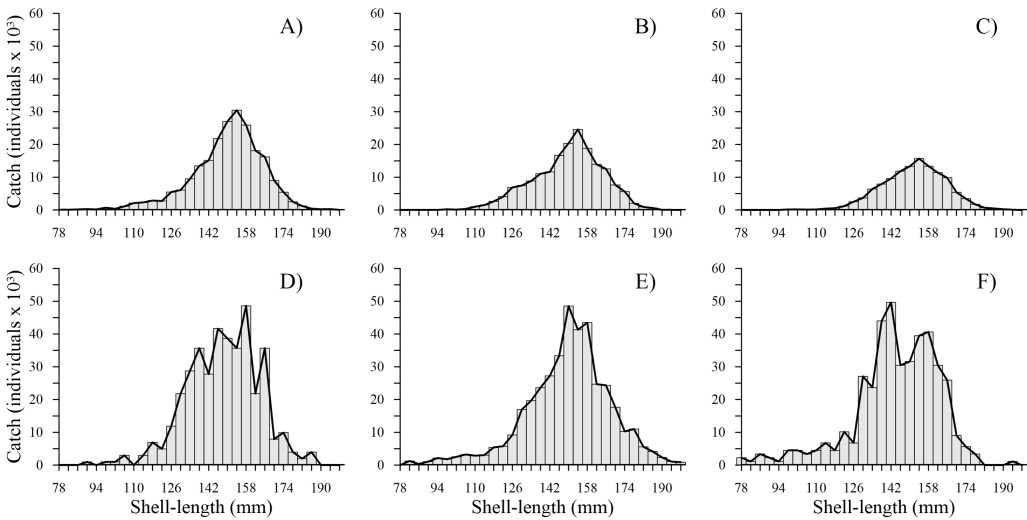

**Figure 2** **Catch-at-shell length analysis for the geoduck clam *Panopea globosa*.** The line is the ICSA model fitted to the observed catch (bars). (A) 2010, (B) 2011, (C) 2012, (D) 2014, (E) 2015, and (F) 2016.

$$TB_l = \sum_{i=1}^{n} N_{l',t'} \omega_l \qquad (6)$$

$$VB_l = \sum_{i=1}^{n} N_{l',t'} \omega_l s_l \qquad (7)$$

where $\omega_l$ was estimated from the power equation $\omega_l = \alpha l^\beta$ (*González-Peláez et al., 2015b*).

## RESULTS

### Shell length structure

The time series showed that from 2010 to 2012, the shell length frequency distributions were similar; the most harvested shell length classes were between 126 and 174 mm, while the least harvested shell length classes were between 178 and 198 mm. From 2014 to 2016, a marked change was observed in the total shell length frequency distribution. We noted an increase in harvest of individuals with a shell length less than 110 mm. Although the shell length classes mostly harvested (118 and 178 mm) and less predominant (182 and 198 mm) were similar to those caught during the first three years, we observed that shell length frequency of geoduck clam was fragmented from 2014–2016, and the bell shape of the shell length frequency observed during 2010–2012 changed to an irregular shape during 2014–2016. During the first three years, the total number of individuals caught per shell length class decreased. In contrast, during the second period, the catches increased along the shell length frequency distribution; thus, two periods were clearly distinguishable (Fig. 2).
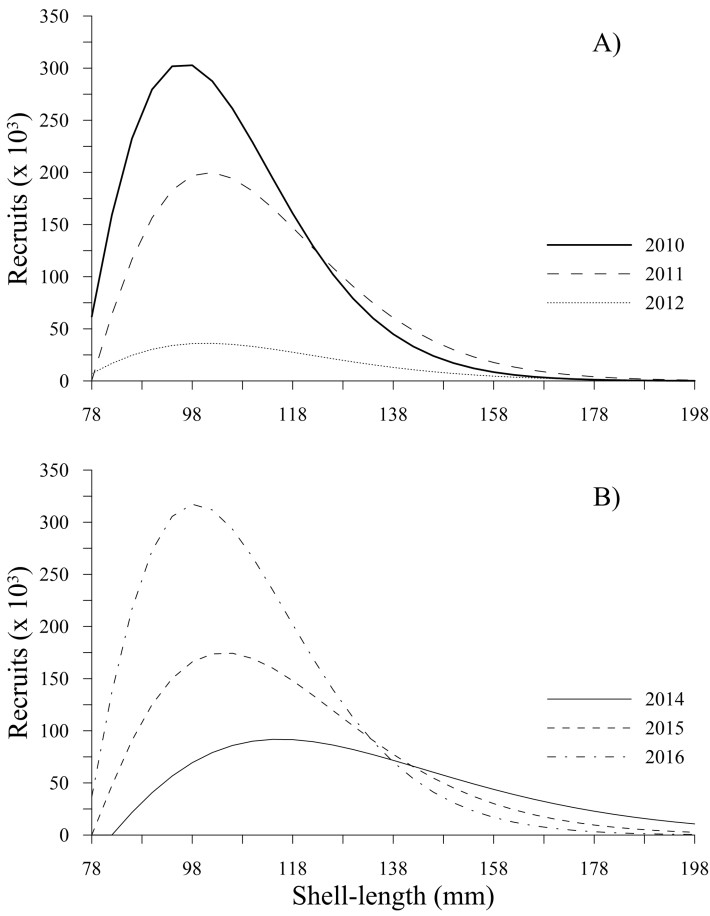

**Figure 3** **Recruitment-at-length estimated for geoduck clam *Panopea globosa* in Puerto Peñasco, Sonora.** (A) Time period from 2010–2012, and (B) time period from 2014–2016.

## Recruitment

The recruitment estimated for *P. globosa* showed a negative trend from 2010 to 2012, with values of $2.78 \times 10^6$, $1.93 \times 10^6$ and $3.75 \times 10^5$, respectively. This sudden decrease to the lowest level represented a change by one order of magnitude in 2012 and a loss of 86% of the recruitment abundance, indicating a decrease over a very short period of time. Subsequently, recruitment increased from $8.85 \times 10^5 (2014)$ to $1.76 \times 10^6$ (2015) and $3.02 \times 10^6$ (2016), reaching recruitment levels slightly greater than those estimated in 2010. Figure 3 shows the changes in number of recruits to the fishery (from 78 to 130 mm) estimated in our study.

## Fishing mortality and selectivity

The fishing mortality estimates calculated by the ICSA model varied over time. The geoducks of 166 mm shell length experienced the main fishing pressure along the time series, exhibiting the highest fishing mortality during 2016 (0.79), followed by 2011 (0.63), 2010 (0.62) and 2012 (0.42). In 2014, fishing mortality showed two peaks at 0.36 in the 158 and 166 mm shell length classes. Finally, during 2015, a peak of 0.22 was observed for the
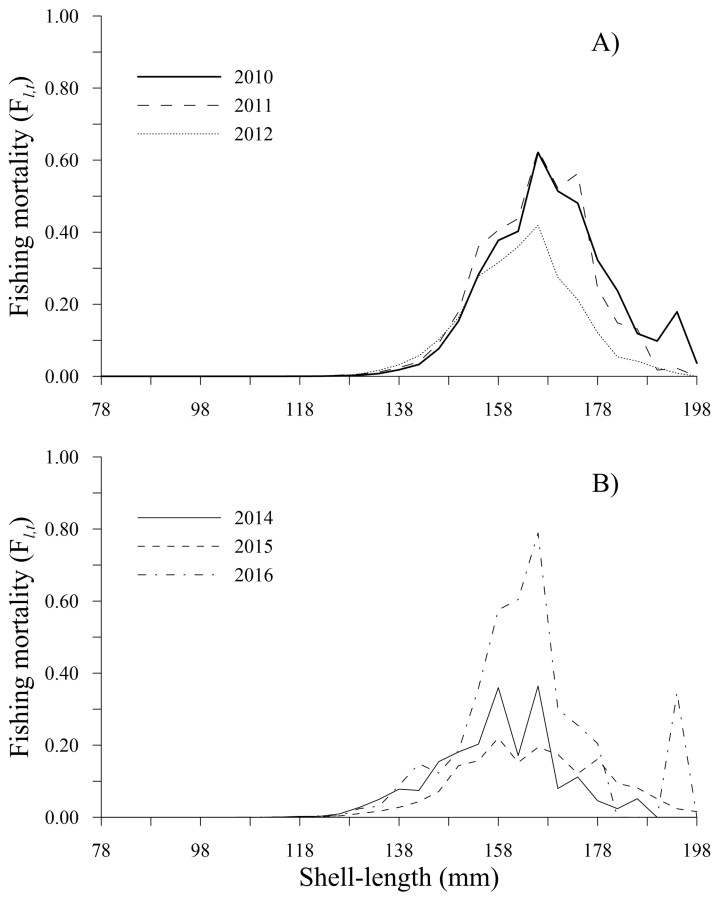

**Figure 4** **Fishing mortality-at-shell length estimated for the geoduck clam *Panopea globosa* in Puerto Peñasco, Sonora.** (A) Time period from 2010–2012, and (B) time period from 2014–2016.

158 mm shell length class. Comparatively, the lowest fishing mortality (<0.05) occurred in individuals in the shell length classes of 78–130 mm (2010–2016) and those larger than 190 mm (2011–2015). Fishing mortality increased in the 194 mm shell length class only during 2010 and 2016 (Figs. 4A, 4B).

The selectivity showed a decrease in shell length from 150 (2012) to 142 mm (2016). This change was observed in the shell length selectivity of geoducks that had a 50% probability of being caught by a diver; this value was defined as $S_{l50\%}$ (Fig. 5). The selectivity estimates for the analyzed years showed the lowest values (<0.07) for the shell length classes between 78 and 118 mm. Changes in selectivity from 2010 to 2015 showed an accelerated increase from 0.2 to 0.99 for individuals with shell lengths between 122 and 174 mm. Comparatively, during 2016, the changes in selectivity included smaller individuals and a wider shell length range (110–174 mm), with values between 0.03 and 0.99. Finally, the geoducks in the shell length classes larger than 178 mm attained the maximum selectivity.

## Management quantities

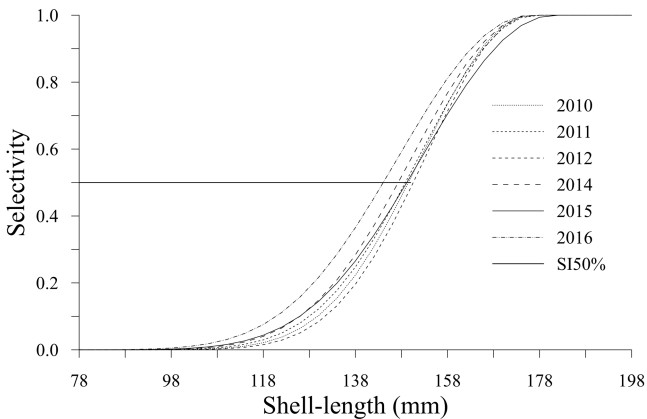

**Figure 5** Selectivity-at-shell length estimated for the geoduck clam *Panopea globosa* in Puerto Peñasco, Sonora.

*Harvest rate*

The tendency of the harvest rates of geoducks between 78 and 115 mm varied slightly throughout the time series, exhibiting values less than 0.05; subsequently, the harvest rate estimates gradually increased for geoducks from 118 to 130 mm, reaching a maximum value of 0.16 (2014) for geoducks of 130 mm (Figs. 6A, 6B). For individuals larger than the minimum legal size, an increasing trend in the harvest rate-at-shell length was also observed. Thus, the individuals between 154 and 178 mm reached maximal values of 0.48 (2010), 0.49 (2011), 0.36 (2012), 0.37 (2014), 0.26 (2015) and 0.56 (2016). A decreasing harvest rate-at-shell length occurred from 2011 to 2015 for the shell length classes between 182 and 198 mm; however, for 2010 and 2016, a patch of large geoducks was located and harvested; consequently, the harvest rate of the 194 mm shell length class suddenly increased. Although the fishing pressure on geoducks, expressed as the harvest rate-at-shell length, was highly variable among the years analyzed, we observed a general pattern over time: the shell length classes between 142 and 174 mm accounted for more than 50% of the harvest (Figs. 7A, 7B).

## Total and vulnerable biomass

The estimates of total biomass-at-shell length showed changes along the time series; from 2010 to 2012, the total biomass-at-shell length decreased; subsequently, an increase was observed during 2014 and 2015, with a slight decrease in 2016 (Table 2). In the first period, the maximum estimates of total biomass were 204.54 t (2010) and 161.10 t (2011) for the shell length class of 122 mm; then, the total biomass-at-shell length rapidly decreased to 65.41 t (2012) for geoducks in the shell length class of 146 mm. During the second period (2014–2016), the highest values of total biomass-at-shell length were 266.66 t (2014), 265.75 t (2015) and 224.36 t (2016) for the shell length classes of 158 mm, 146 and 126 mm, respectively (Figs. 8A, 8B). The shell length ranges most representative of geoduck clams associated with the total biomass were different for each time period analyzed. For the first period, the range of shell length changed from 110-138 mm (2010–2011) to 130–170

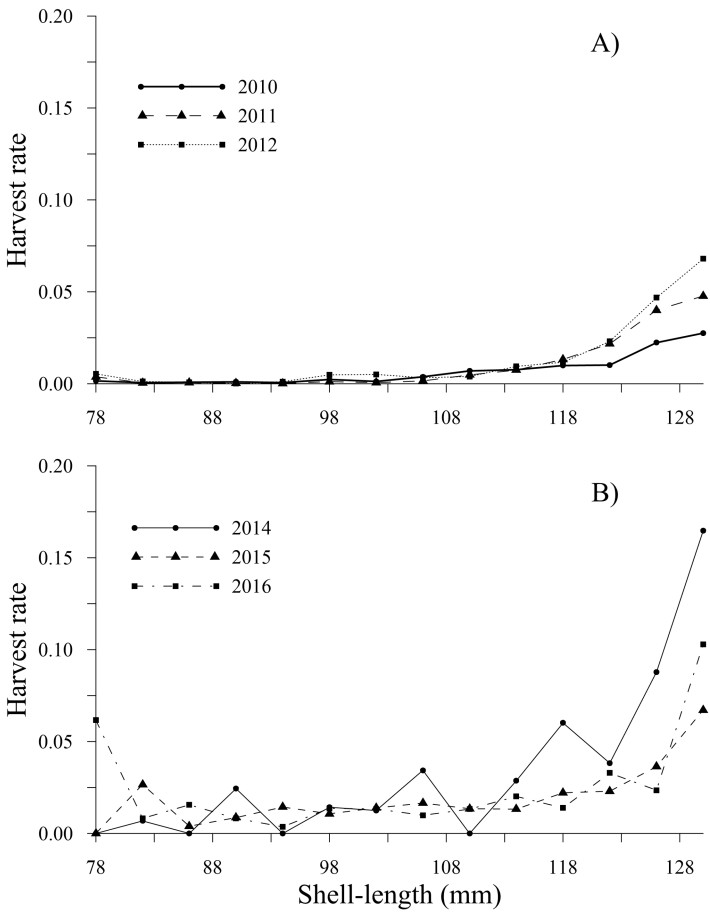

**Figure 6  Harvest rate-at-shell length estimated for the geoduck clam *Panopea globosa* smaller than the minimum legal shell length of 130 mm.** (A) Time period from 2010–2012, and (B) time period from 2014–2016.

mm (2012). This change indicated that the total biomass of younger individuals decreased, and during the third year, although the individuals were larger, their total biomass was diminished, representing the year with the lowest biomass. During the second period, the range of shell length progressively diminished, and the ranges among years were 138-170 mm (2014), 130–138 mm (2015), and 110–142 mm (2016) (Figs. 8A, 8B).

The time series showed that the vulnerable biomass-at-shell length decreased from 2010 to 2012; subsequently, during 2014 and 2015, the vulnerable biomass exhibited an increase, diminishing in 2016 (Table 2). During the first period, the peaks of vulnerable biomass were 72.93 t (2010) and 52.77 t (2011) for the 154 mm shell length class and 43.62 t during 2012 for individuals with a shell length of 162 mm. Comparatively, during the second period, the highest values of vulnerable biomass were 237.90 t (2014), 170.66 t (2015), and 99.72 t (2016) for the 178, 162 and 150 mm shell length classes, respectively. In the time series, vulnerable biomasses less than 25 t were observed for individuals between 78 and 110 mm and larger than 186 mm, excluding 2014, when the vulnerable biomass for

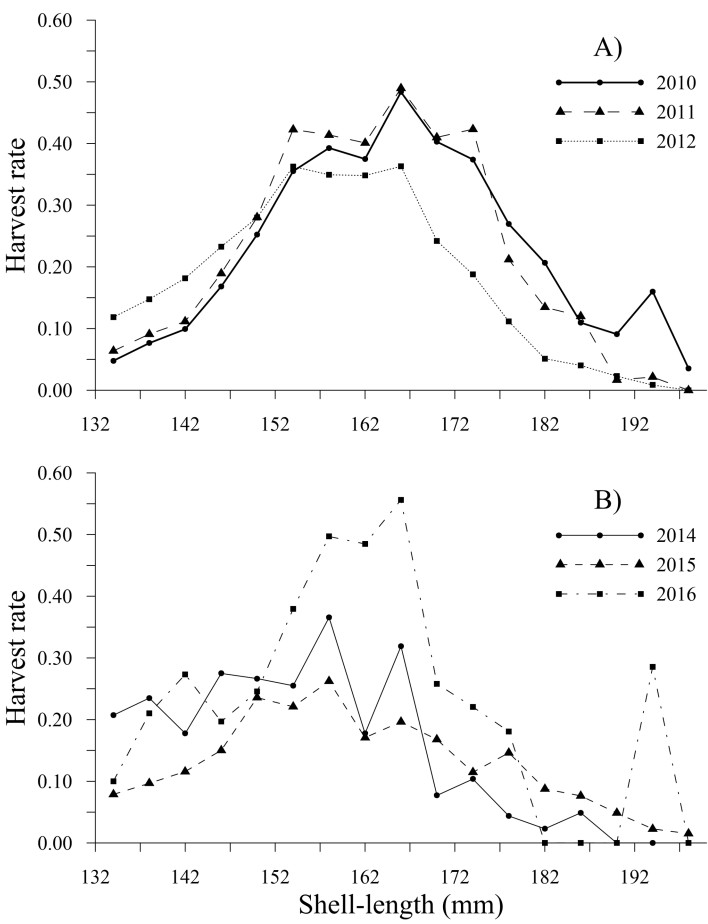

**Figure 7** **Harvest rate-at-shell length estimated for the geoduck clam *Panopea globosa* larger than minimum legal shell length of 130 mm.** (A) Time period from 2010–2012, and (B) time period from 2014–2016.

**Table 2** **Management quantities estimated for the population of the geoduck clam *Panopea globosa* in Puerto Peñasco, Sonora.**

| Management quantities | 2010 | 2011 | 2012 | 2014 | 2015 | 2016 |
|---|---|---|---|---|---|---|
| Total abundance ($N \times 10^6$) | 4.63 | 3.01 | 1.03 | 2.88 | 4.72 | 5.15 |
| Total biomass (t) | 3,262 | 2,517 | 1,130 | 4,801 | 4,870 | 3,619 |
| Vulnerable biomass (t) | 672 | 555 | 480 | 2,820 | 2,189 | 1,104 |

individuals between 190 and 198 mm was not estimated because these shell length classes were not harvested. Conversely, during 2015, the vulnerable biomass for the individuals in the 198 mm shell length class showed a slight increase (Figs. 9A, 9B).

## Parameter estimation

The parameters associated with the population dynamics of the geoduck clam for each year of fishing analyzed are shown in Table 3. The interaction between the different modules of the ICSA model exhibited convergence in the objective function. Thus, the ICSA model
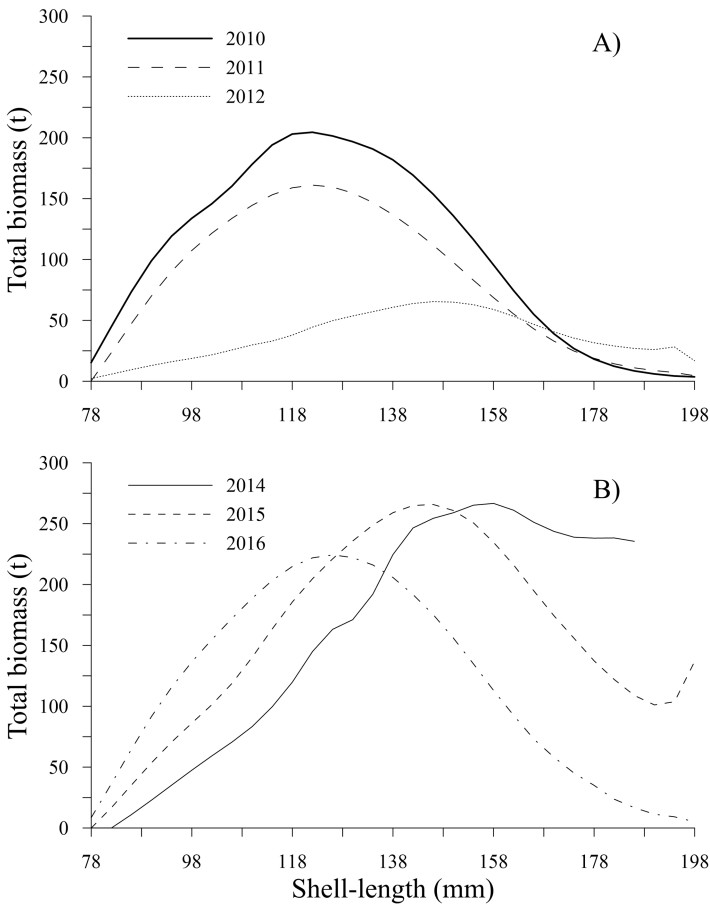

**Figure 8  Total biomass-at-shell length estimated for the geoduck clam *Panopea globosa* in Puerto Peñasco, Sonora.** (A) Time period from 2010–2012, and (B) time period from 2014–2016.

showed high performance in fitting the observed catch-at-shell length data, exhibiting low residual values (*RSS*) (Table 3).

## DISCUSSION

The population dynamics of *Panopea* spp. are very different from those of r-selected species; geoduck clam exhibits a longevity of 34 years in the study area (*Aragón-Noriega, Calderon-Aguilera & Pérez-Valencia, 2015*), late maturity, a low individual growth rate, and a high natural mortality rate for early stages and young individuals (*González-Peláez et al., 2015a*). These demographic traits make to geoduck susceptible to fishing mortality, negatively affecting its abundance, spatial density, survival and reproduction rates; hence, stock management must avoid significant decreases in abundance. Therefore, the correct interpretation of our results has a biological component associated with geoduck clam fishery management, where the variability in recruitment and biomass are important to the population dynamics of the geoduck clam.

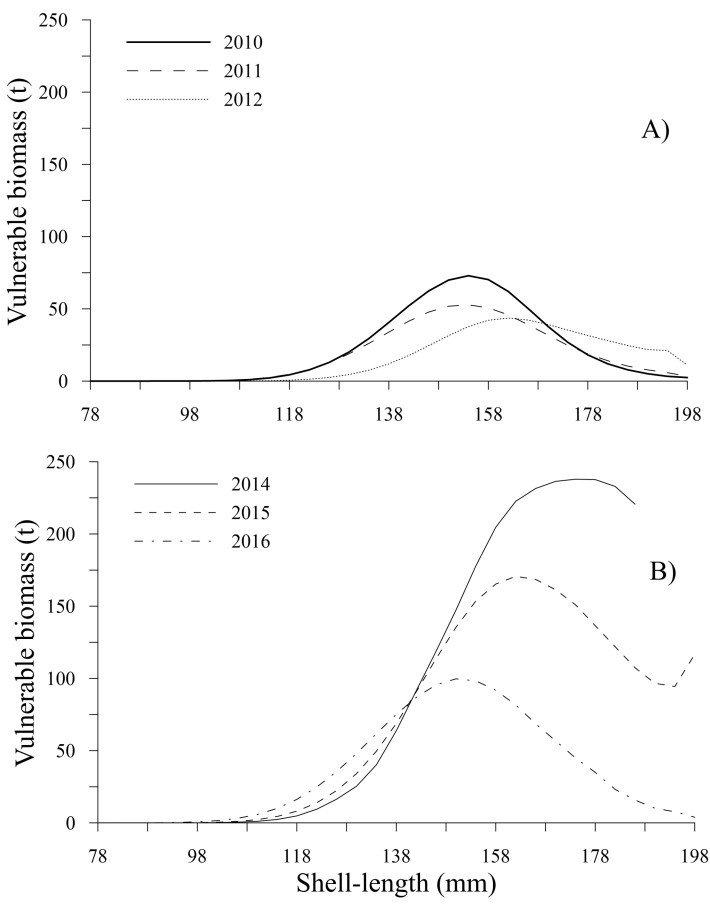

**Figure 9 Vulnerable biomass-at-shell length estimated for the geoduck clam *Panopea globosa* in Puerto Peñasco, Sonora.** (A) Time period from 2010–2012, and (B) time period from 2014–2016.

**Table 3 Parameters and *RSS* estimated by the ICSA model for the geoduck clam *Panopea globosa*.**

| Parameters | 2010 | 2011 | 2012 | 2014 | 2015 | 2016 |
|---|---|---|---|---|---|---|
| $\hat{N}_{l,0} \times 10^6$ | 1.81 | 0.83 | 0.69 | 1.68 | 2.82 | 2.03 |
| $R_t \times 10^6$ | 20.53 | 22.12 | 6.04 | 41.63 | 31.05 | 28.12 |
| $\alpha_r$ | 14.97 | 12.11 | 9.64 | 6.48 | 9.69 | 13.27 |
| $\beta_r$ | 5.26 | 6.68 | 8.12 | 13.11 | 8.38 | 6.00 |
| $\alpha_s$ | 56.23 | 49.92 | 60.78 | 46.46 | 40.10 | 37.61 |
| $\beta_s$ | 3.16 | 3.58 | 2.93 | 3.83 | 4.56 | 4.73 |
| $\beta_g$ | 0.20 | 0.50 | 0.15 | 0.13 | 0.31 | 0.61 |
| *RSS* | 6.15 | 6.78 | 4.07 | 4.12 | 0.05 | 1.54 |

The results showed that individuals smaller than the minimum legal size (130 mm) were harvested, even the harvest rate-at-shell length increased after 2010 and these clams are not targets of the geoduck clam fishery. Moreover, individuals with shell lengths larger than 130 mm were also excessively harvested, with a harvest rate-at-shell length greater

than 1%, denoting that the shell length structure of geoduck clam was overfished over time. Although a minimum legal size of 130 mm was established as a management tactic to avoid growth overfishing, it failed to maintain the sustainability of the geoduck clam stock in Puerto Peñasco, Sonora. Thus, two overfishing patterns were simultaneously identified: (a) growth overfishing (which occurs when young clams are caught before reaching their optimal size and the fishery cannot produce the maximum yield) and (b) recruitment overfishing (which represents excessive fishing pressure on the spawning stock, reducing its abundance to a very low level, at which the spawning stock cannot produce enough new individuals for recruitment).

According to *Hilborn & Walters (1992)*, a rapid depletion and potential collapse occur in commercial fisheries when failures in recruitment are observed. In this study, recruitment to the fishery was assumed to occur over a range of shell length classes (78–130 mm); thus, a rapid decrease in recruitment was observed, representing an 86% in the overall recruitment estimated from 2010 to 2012. The sudden decrease in recruitment occurred after two years of excessive harvesting of all sizes of *P. globosa*. Usually, the variability in recruitment of *Panopea* spp. can be explained by the following: a) the failures in recruitment, which have been documented even in years where there was not exploitation, thus the decrease is not attributable to harvest (*Zhang & Campbell, 2004*); (b) a loss of recruits, which has been identified in populations with low densities of individuals (*Campbell et al., 2004*); (c) synchronic variability in recruitment, which has been corroborated by the comparison of fishing and nonfishing areas (*Campbell et al., 2004*); (d) negative effects of fishing mortality on recruitment, identifiable on a short-term scale (*Campbell et al., 2004*); and (e) declining and rebounding trends in recruitment, which have been identified as consequences of environmental variables (*Valero et al., 2004*).

Additionally, for *Panopea* spp., failures in recruitment have been identified to occur when a unimodal distribution pattern in the shell length structure is observed; this pattern is mainly caused by the presence of old individuals and an evident lack of young individuals, indicating poor recruitment pulses (*González-Peláez et al., 2015b*). A low frequency of young individuals can also be influenced by the spatial heterogeneity typical of *Panopea* spp. (patchy distribution), where the recruits of a cohort can be located (settled and recruited) in different beds due to the process of advection or the retention of larvae. As was reported for *P. globosa* in the upper Gulf of California, the geoduck clam beds from Puerto Peñasco and Guaymas (east coast of the Gulf of California) are influenced by the supplies of geoduck clam larvae from the San Felipe beds (west coast of the Gulf of California) due to anticyclonic circulation during the spawning period in the winter (*Munguia-Vega et al., 2015*). In addition, for geoduck clam fishery, the removal of adults may reduce the reproductive output of the population.

Although this study is focused on the population dynamics of the geoduck clam and the estimation of management quantities (e.g., biomasses), the changes in biomass and recruitment could be explained by the management scheme currently applied to this fishery; the rationale is based on the historical development of the geoduck clam fishery in the region. In Mexico, the stock assessment of geoduck clam was initially based on data poor, and the management rules originally proposed had the following limitations: (1)

uncertainty in biomass estimates and a lack of knowledge regarding biological reference points, (2) failure to identify warning signs of depletion, and (3) poor management outcomes. Although specific fishing management plans by species currently exist, these plans are not supported by biological and demographic bases of *Panopea globosa* and *P. generosa*; therefore, the regulations are indiscriminate for both species and they have been managed as a single species (*Aragón-Noriega et al., 2012*). The harvest rates of 0.5% applied during the predevelopment phase (2008) and 1% during the growth phase (2012) of the geoduck clam fishery were adequately implemented during the first years as a result of the management rules proposed by the Mexican government; however, the data and biological knowledge were insufficient at that time. Consequently, the harvest policies were not evaluated or updated in the short term, and passive management measures, such as regulations based on the control of fishing effort, gear restrictions, and minimum harvest size, failed to maintain stable geoduck biomass levels.

According to *Zhang & Hand (2006)*, a decay cohort model for geoduck can be used for modeling the effect of a given exploitation rate; the assumption shows that if the population is close to the virgin condition, then a management strategy of constant harvest rate can be implemented into a defined time scale. *Zhang & Hand (2006)* performed a simulation projected forward for 50 years by varying the exploitation rate from 0% to 4% using an increment of 0.5%. Thus, the reference point would be the 50% of the virgin biomass for a specific fishing ground; evidently if the exploitation rate is increased and maintained constant over time, then the reference point is reached in less time. This pattern indicates that although a harvest rate can be successfully implemented, the time during which the fishing pressure can be applied is also important. For geoducks in the study area, there are no estimates of population declines over time. The fishery began using a harvest rate of 0.5%, which was later modified to 1%. However, the use of this harvest rate did not allow the determination of whether the population was close to its virgin biomass; moreover, the addition of new beds and expansion of the fishing ground hide the effect of fishing on the geoduck population. Ideally, the harvest rate of 1% is adequate, but the implementation requires a time scale, which was fixed in 50-year horizon (*Aragón-Noriega et al., 2012*). However, this time horizon is based on geoduck clams distributed in northern latitudes (USA and Canada), where the longevity of this species has been estimated to exceed 100 years (*Hoffman, Bradbury & Goodwin, 2000*; *Bureau et al., 2002*; *Bureau et al., 2003*). In the study area, the species has exhibited longevity of 34 years (*Pérez-Valencia & Aragón-Noriega, 2013*; *Aragón-Noriega, Calderon-Aguilera & Pérez-Valencia, 2015*); therefore, the time horizon associated with the harvest rate should be shorter. According to *Zhang & Hand (2006)* the ratio ($\tau$) of current biomass ($B_t$) to virgin biomass ($B_0$) has the following relationships:

i) $\tau = \frac{B_t}{B_0} = 0.5$; the value represents the biological reference point.

ii) $\tau = \frac{B_t}{B_0} < 0.5$; recovery actions are necessary in the fishery.

iii) $\tau = \frac{B_t}{B_0} > 0.5$; the harvest is allowed.

Given that the lifespan of geoducks is limited to 34 years, a $\tau$ in which $B_0$ includes a time horizon of 50 years is not possible because the $B_0$ under a constant harvest of 1% every

year will cause total depletion of the virgin biomass sooner than 50 years. According to this rationale, the constant harvest of 1% for geoducks in the study area should be ideally applied over a 17-year horizon. Therefore, an indicator of sustainability is not limited the allowable harvest rate; instead, the adequate indicator must be a specific $\tau$ value for each fishing ground, including a correct time horizon.

Additionally, the regulation based on a minimum legal size assumes that all the individuals in the shell length of 130 mm reached maturity and had at least one reproductive event (*DOF, 2012*). However, for populations of *P. globosa* from Guaymas and San Felipe in the Gulf of California, a size-at-maturity at 50% of 88.75 mm and 89.37 mm shell length was determined, respectively; these values were 41 mm below the minimum legal size established for the geoduck fishery (*Aragón-Noriega, 2015*). Comparatively, the size-at-maturity at 50% estimated for *P. globosa* was higher than those reported for *P. generosa* in Washington (75 mm) and two populations in Canada: Gabriola Island (58.3 mm) and Yellow Bank (60.5 mm), as well as for two populations of *P. zelandica* from New Zealand: Kennedy Bay (55 mm) and Shelly Bay (57 mm) (*Andersen Jr, 1971*; *Campbell & Ming, 2003*; *Gribben & Creese, 2003*). These differences in the maturity size among populations of *P. globosa* should be taken into account when establishing management strategies, mainly because the maturity size at 50% is positively correlated with its longevity; therefore, individuals mature at a specific fraction of their maximum length (*Jensen, 1996*; (*Law, 2000*; *Nadon & Ault, 2016*). This aspect is relevant for *P. globosa*, its individual growth is highly variable; in Bahía Magdalena (Pacific coast), geoduck clams exhibited an asymptotic shell length of 179.85 mm (*Luquin-Covarrubias et al., 2016a*; *Luquin-Covarrubias et al., 2016b*); the geoduck population from Guaymas (Central Gulf of California) had an asymptotic length of 122 mm (*Cortez-Lucero et al., 2011*; *Cruz-Vázquez et al., 2012*), while in Puerto Peñasco and San Felipe (Upper Gulf of California), the asymptotic length values were 161.79 mm and 190.84 mm, respectively (*Aragón-Noriega, Calderon-Aguilera & Pérez-Valencia, 2015*).

Therefore, the minimum legal size of 130 mm is not a useful management rule because it is not supported by the specific life history traits of *Panopea globosa* in each region. According to *Muse (1998)*, the minimum size limit is not a practical management tool for geoduck clams, and the economic or management overvaluations can be a serious problem. The main reason is that the mortality rate of discarded geoducks may be 100% because the shell length cannot be known until the individual is extracted from the marine substrate. Additionally, geoducks are unable to completely close their shells; thus, once an individual is removed, it cannot rebury itself, leaving it vulnerable to depredation and dying within a short period of time. Given that the individuals to be harvested are located through their siphon holes on the substrate, attempts to determine the shell length through hole size or the width between paired siphon holes were documented for *P. generosa* and *P. zelandica*, but these methods proved to be poor predictors of shell length (*Andersen Jr, 1971*; *Gribben & Creese, 2003*). Thus, *Orensanz et al. (2004)* explained that several factors influence the vulnerability of geoduck clams, such as the experience of the divers and their ability to identify beds with high densities or high numbers of siphons, which varies according to type of substrate and affects harvest success. Initially in the study area, small clams were separated from the harvest and fishers tried to rebury them in shoreline areas, but this practice did not

succeed because the geoducks did not remain buried in the marine substrate. Eventually, geoducks from the 100 mm shell length had a high market price, and their harvest began to be a profitable activity (*Aragón-Noriega, 2015*). Consequently, geoducks smaller than minimum legal size were harvested in Puerto Peñasco; however, these individuals were not reported, affecting the quantification of harvest rate legally established at 0.5% and 1%. Potentially, the harvest could be greater, causing a rapid depletion of the fishing ground as indicated by the dramatic changes in harvest rate-at-shell length estimated in this study.

Geoduck clams, *Panopea globosa,* are spatially structured as "meta-populations," in which benthic subpopulations are connected with each other through the dispersal of pelagic larvae. Within populations, individuals are patchily distributed. According to *Orensanz & Parma (2016)*, a fishing ground is typically occupied by a meta-population. Beds within a fishing ground are more or less discrete areas with high density. Finally, within beds, individuals are distributed and concentrated in patches of relatively high density. Thus, the results obtained from the ICSA model represent the changes in the population dynamics at length from only one population of *P. globosa* as well as from different fishing beds within the population. According to technical reports published by the National Fishery and Aquaculture Institute from Mexico (governmental agency for fishery management) (*Ochoa-Araiza et al., 2014*; *Ochoa-Araiza et al., 2015*), we observed spatiotemporal changes in the harvestable area; (a) from 2009 to 2012 (fishery predevelopment phase), small patches were initially identified and harvested, mainly into shoreline areas; and (b) from 2013 to 2016 (fishery growth phase), several patches were added, and the fishing area was expanded; thus, the harvest of geoduck clams progressed offshore. Spatially, from 2009 to 2016, the fishing area increased in size toward the southeast of the study area, indicating the greater expansion of the harvestable area in the region. Comparatively, the fishing zone in the northeast area suffered fragmentation, which means that the initially harvested patches showed spatial reductions such that only a small area remained (Fig. 10, Appendix A1). The spatial changes had negative impacts for several benthic populations; for example, the *Haliotis* spp. stocks suffered a dramatic serial depletion, both spatially and by species, and were gradually replaced by other abalone species until their decline (*Karpov et al., 2000*; *McCormick, 2000*; *Morales-Bojórquez, Muciño Díaz & Vélez-Barajas, 2008*). Another example is the Catarina scallop (*Argopecten circularis*), which was exploited in shallow beds; when the landings attained the maximum levels, the areas were overfished, and the divers progressively moved their fishing grounds to deeper beds (*Maeda-Martínez et al., 1993*; *Rothschild et al., 1994*; *Tracey & Lyle, 2011*). In this study, changes in the total and vulnerable biomass of geoduck clam occurred in a short period time, indicating a decrease from 2010 to 2012 and an increase from 2014 to 2016, which cannot be explicitly explained by the population dynamics of long-lived species such as *P. globosa*. Recently, these declines and recoveries of recruitment and biomass (total and vulnerable) in a short time were also observed in the population of *P. globosa* in Bahia Magdalena, where they were associated with an effect of serial depletion (*Amezcua-Castro et al., 2019*).

According to the bathymetry around Puerto Peñasco, the geoduck fishing beds are located at depths between 2 and 30 m, but the commercial harvest has gradually moved

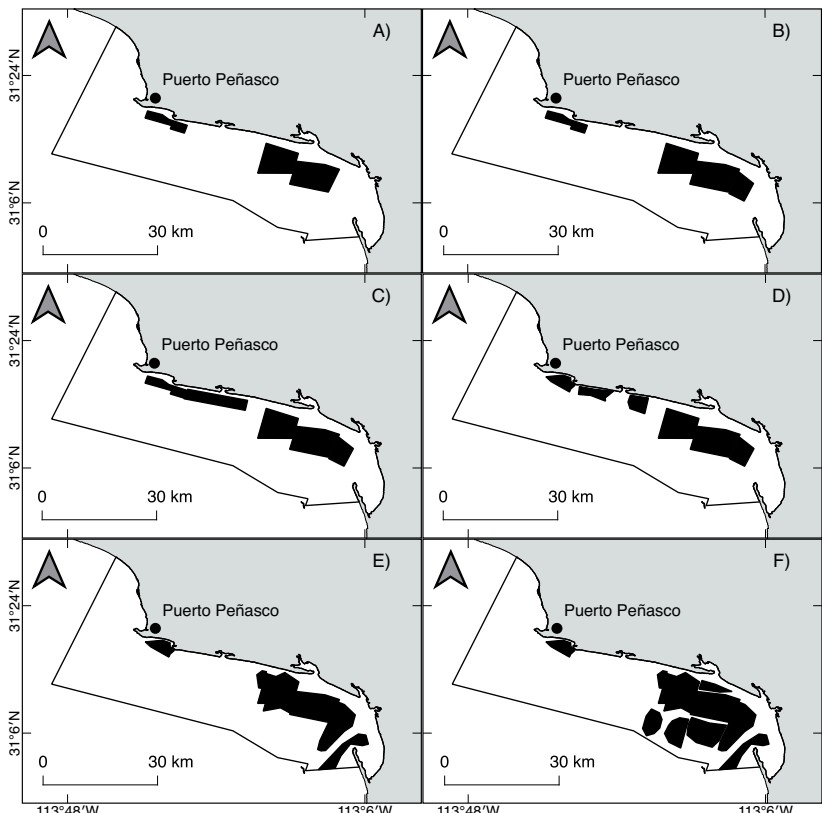

**Figure 10  Spatiotemporal changes in the fishing area of the geoduck clam *Panopea globosa* from 2009 to 2016.**  To find the coverage of a fishing area, the station positions were loaded into GIS software; the boundary stations were traced, and the enclosed areas were calculated (km²) (*Ochoa-Araiza et al., 2014*; *Ochoa-Araiza et al., 2015*). (A) 2009, (B) 2010, (C) 2011, (D) 2012, (E) 2013, and (F) 2014–2016. The figure was created and edited by Daniela Maldonado Enriquez (CIBNOR SC).

from the shoreline (9 m) to offshore areas (30 m). For geoduck clam, factors such as the depth, current flow, substrate composition, and geographic area have been shown to affect the shell length of individuals in any particular bed (*Goodwin & Pease, 1991*). These authors also showed significant differences in the shell length of geoducks at several depths, indicating a decrease in shell length from 146.2 mm in shallow areas (9.1 m) to 125.8 mm in deeper waters (13.7 m). Our results showed a change in the shell length frequency distribution during the second period (2014–2016), representing a high harvest of individuals smaller than 110 mm as well as a reduction of eight mm in the $S_{l50\%}$, these changes coincided with the movement of fishing activity to deep waters (30 m), where different conditions than those of shallow water can influence the growth of geoducks. Geoducks live buried up to 1 m deep within sand and mud substrates, and they feed by filtering phytoplankton from sea water (*Goodwin & Pease, 1989*); thus, in deep areas where the current speed is typically lower, the deposition of fine sediment affects the food availability and therefore reduces the growth of the geoducks to smaller lengths in comparison with those living in high-current areas (*Goodwin & Pease, 1991*). In this way,

given the changes in the fishing area and the relocation of fishing mortality, a spatial serial depletion could be operating in this fishery (*Botsford, Campbell & Miller, 2004*; *Orensanz et al., 1998*; *Kirby, 2004*). Fishers are moving the fishing effort along the latitudinal gradient and from shoreline to offshore areas; therefore, the biomass and recruitment cannot be adequately quantified because new beds containing an abundance of younger *P. globosa* individuals may be available for harvest. For this reason, the results obtained from the ICSA model showed an increasing trend in the recruitment and biomass estimates; however, both increases may be masked due to serial depletion.

Therefore, some type of area management is necessary for this sedentary species to maximize the yield, as has been established for the *Panopea generosa* fishery in British Columbian waters, where a "three-year rotation time" is maintained within three regions (north, central, and south) and each region is divided into three subregions, with one out of three subregions within each region harvested every 3 years. For Washington, USA, a preharvest survey is conducted in each tract to estimate the biomass available for the geoduck fishery; if this biomass is high enough, then the harvest is authorized. A depleted tract cannot be opened to the harvest until a survey indicates that it has recovered to preharvest conditions. The geoduck management uses a *de facto* rotation, where the fast-recovery tracts would be revisited more often than slow-recovery ones (*Orensanz et al., 2004*). For Mexican waters, although area rotations are considered in the fishery management plan, this practice is not adequately implemented. Rotational fishing is part of a precautionary approach, and the Mexican geoduck clam fishery needs to analyze the possibility of managing high- and low-productivity regions separately or to use an area rotation system to prevent possible declines in the different populations. Explicitly, two options for area management should be analyzed: (a) pulse rotation (high levels of effort applied in an area when it is opened, and all exploitable organisms are harvested at periodic intervals); and (b) symmetric rotation (which requires less concentrated effort and allows areas to be open half the time). The advantage of rotational fishing is that it prevents the impact of both growth and recruitment overfishing (*Myers, Fuller & Kehler, 2000*; *Hart, 2001*; *Hart, 2003*; *Harris, Adams & Stokesbury, 2018*).

## CONCLUSIONS

The results obtained in this study indicated overfishing of the geoduck clam population in Puerto Peñasco, Sonora, in the upper Gulf of California, which had not been identified in the stock assessment previously proposed by the fishery management plan for *P. globosa* (*DOF, 2012*). The geoduck clam fishery in Mexico continues to be regulated through a passive management scheme, which is unable to identify the variability in the total biomass or changes in recruitment over time. The ICSA model used in this study was able to evaluate individual growth patterns as well as recruitment, selectivity, harvest rates, and survival rates. Thus, the population dynamics is included in the final estimations of the management quantities, such as biomass-at-shell length (total and vulnerable) and harvest rate-at-shell length. The harvest rate and variability in recruitment between beds have important implications for geoduck clam management and should be considered

to avoid serial overfishing, particularly if the populations are structured in small isolated beds, which increases the possibility of overfishing. Therefore, geoduck clam management should include a conservative approach for each fishing area, improving both conservation and fishery objectives to avoid the depletion of *P. globosa* population.

## ACKNOWLEDGEMENTS

The authors thank the Coops Mar y Tierra del Golfo de Cortéz, Buzos de Puerto Punta Peñasco, Jaiberos y Escameros, and Islas de Sonora for the logistic and field support. Thanks to Daniela Maldonado Enriquez for her support in image editing.

### Funding

Marlene Anaid Luquin Covarrubias was supported by Consejo Nacional de Ciencia y Tecnología México (CONACyT) for the PhD fellowships CVU 636852/337484. The funders had no role in study design, data collection and analysis, decision to publish, or preparation of the manuscript.

### Grant Disclosures

The following grant information was disclosed by the authors:
Consejo Nacional de Ciencia y Tecnología México (CONACyT) for the PhD fellowships CVU 636852/337484.

### Competing Interests

The authors declare there are no competing interests.

### Author Contributions

- Marlene A. Luquin-Covarrubias conceived and designed the experiments, performed the experiments, analyzed the data, prepared figures and/or tables, authored or reviewed drafts of the paper, and approved the final draft.
- Enrique Morales-Bojórquez conceived and designed the experiments, analyzed the data, authored or reviewed drafts of the paper, and approved the final draft.
- Juan A. García-Borbón and Sergio Amezcua-Castro analyzed the data, authored or reviewed drafts of the paper, and approved the final draft.
- Sergio A. Pérez-Valencia and Estefani Larios-Castro performed the experiments, authored or reviewed drafts of the paper, and approved the final draft.

### Animal Ethics

The following information was supplied relating to ethical approvals (i.e., approving body and any reference numbers):

The Comisión Nacional de Acuacultura y Pesca (CONAPESCA) approved this study [a) DGOPA.05338.050710.3326 (2010), b) DGOPA.09450.221111.3442 (2011), c) 126054025018 (2012), d) PPF/DGOPA-186/14 (2014), e)126070025019 (2015), and f) 126047025017 (2016)].

### Field Study Permissions

The following information was supplied relating to field study approvals (i.e., approving body and any reference numbers):

The annual catch data were obtained from the Subdelegación Federal de Pesca at Puerto Peñasco, Sonora, Mexico (Comisión Nacional de Acuacultura y Pesca, CONAPESCA). Permission to use these commercial and biological fisheries data was granted by the Mexican federal government (CONAPESCA) through scientific collecting permit numbers DGOPA.05338.050710.3326 (2010), DGOPA.09450.221111.3442 (2011), 126054025018 (2012), PPF/DGOPA-186/14 (2014), 126070025019 (2015), and 126047025017 (2016) with fishermen of the SCPP Buzos de Puerto Peñasco, SCPAP Islas de Sonora, SCPP Jaiberos y Escameros, and SCPP Mar y Tierra del Golfo de Cortez.

### Data Availability

Raw data are available in Data S1.

### Supplemental Information

Supplemental information for this article can be found online at http://dx.doi.org/10.7717/peerj.9069#supplemental-information.

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
