# Peer review of "Evidence of overfishing of geoduck clam Panopea globosa from a length-based stock assessment approach"

_PeerJ, doi:10.7717/peerj.9069_

## Round 0.1 · original submission · Major Revisions

The paper should be improved. Particularly the methods the explanation and the scope of the results should be more developed. Please follow all the indications of the reviewers before forwarding the paper for your new evaluation

·

Basic reporting

a. Abstract line 38 – subject-verb agreement “the dynamics were (not was)” here and in other usage in the manuscript
b. Introduction line 79 - I assume that P. globosa are unable to survive initial excavation. Since the authors raise a minimum legal size here, I think it is important background information to explain how a minimum legal size would work if size cannot be determined prior to excavation.
c. Methods line 171 – I don’t understand what is meant by a “gnomonic interval”. The shortest amount of time possible to complete the stage?
d. General – can you make clear that the fishery (harvest) continued in 2013, even if the sampling program did not?
e. Results line 321 – I usually define a “harvest rate” as the amount harvested divided by the total biomass. It appears that you are reporting here a proportion that each size class represents in the total harvest amount. Or do you estimate you harvested up to half of an entire size class in one year? Please clarify.
f. Discussion line 359 – This sentence could be revised for clarity. It is unclear whether the second half of the sentence refers to geoduck or shrimp/squid.
g. Figure 6 – The reference lines in this figure correspond to 10% and 5% harvest rates. The introduction (line 80) refers to 1% and 0.5% harvest rates. Is one of them a typo?

Experimental design

a. Methods line 144 – Can you give more detail about how sampling was randomized? Were the commercial fishers taking samples at the direction of the authors, or were the authors sampling from regular commercial harvest? Either way, how were locations, depths, etc. managed to ensure a random sample of the population and not one that follows the harvest choices of the fleet?
b. Methods line 222 – I am confused by how the data collected were translated into measures of recruitment. Were many smaller individuals sampled in the field each year, or was recruitment back-calculated from adult growth and survival in the model, or was it reconstructed from aging? If recruitment to the fishery occurs over a range of size classes in the model, isn’t that the same as fishery selectivity of those size classes? In certain size classes, are some individuals assumed to be recruits and others not? Perhaps you can add explanation in plain language to this section to help me understand.
c. Methods line 238 – I assume GRG stands for generalized reduced gradient (although please spell out all acronyms). I am not familiar with this method, and so my ignorance may be betrayed by my question. By using a non-linear fit in phases, is there a risk to “over-fitting” the model – so that RSS criterion is minimized but generalizability is also low? I am concerned that over-fitting could lead to large fluctuations in the output, which seems to be the case in your results.

Validity of the findings

a. Results line 292 – I’m curious if any external validation or “ground-truthing” can be done for the estimates of biomass over time. Do you have any guess for the prefishing biomass for this area? Do the results reported (Table 3) correspond to harvest removed (2010 – 2012) and the natural mortality rate? I’m guessing they do not, given that the target harvest rates (introduction line 80) and natural mortality estimates reported are quite low, while the biomass is reduced by two-thirds in two years. Similarly, the biomass quadruples in two later years (2012-2014). That does not seem possible for a species like P. globosa – which casts doubt onto the methods used to estimate biomass. Or perhaps this expansion in biomass is due to additional harvest areas being identified?
b. Discussion line 373 – Unfortunately I cannot evaluate the recruitment results because of my inability to understand how they were calculated. However, I enjoy the general discussion around recruitment in this genus.
c. Discussion line 399 – I think this discussion could benefit from a clearer link between the results (harvest rate at shell length) and the conclusion that overfishing is occurring. What harvest rates at shell length would be indicative of sustainable harvest?
d. Discussion line 410 – I don’t understand how a size limit is used in fishery management given that harvesters cannot tell size before excavation.
e. Discussion line 453 - Given the variability in results, particularly in biomass, this line is particularly troubling “If fishers are moving the fishing effort along the latitudinal gradient and from shoreline to offshore areas, then the biomass and recruitment cannot be adequately quantified…”. I agree, and serial depletion is certainly a concern. Are fishers moving the effort or not? In particular, are the results from 2010-2012 and 2014-16 potentially different populations? If they are, does this not invalidate much of your attempt to estimate biomass using length frequencies? It seems from Figure 9 that harvest is shifting in space, but the discussion seems uncertain on this point. I also assume that moving off shore results in a deeper depth of harvest, which could have important implications, but I am not familiar with the bathymetry around Puerto Peñasco.
f. Discussion line 462 – I like this idea of managing the high- and low-productivity regions separately.

Additional comments

This is an interesting and important study describing an attempt to detect overfishing in a data-limited, emerging fishery. I have concerns about the appropriate use of a length-based approach in a species that is long-lived and with slow and variable growth. Data on P. generosa show highly variable size with age, and I don't see evidence here that P. globosa would be much different. The length-distribution methods appear to detect large fluctuations in population biomass, which seem unlikely given the species’ life-history. A possible explanation for the results, a shift in harvest areas, is particularly concerning. It suggests that the population was not sampled randomly (although description of the methods is insufficient to tell) and that the methods may not be a good match for the conclusions. More detail and clearer explanation of the methods and their link to the conclusions will aid in evaluating this. There is certainly potential evidence that this population is being overfished or serially-depleted, and as such this could be an important contribution to science and fishery management. However, the manuscript could be re-worked substantially in order to make a clear case.

Reviewer 2 ·

Basic reporting

no comment

Experimental design

no comment

Validity of the findings

The interpretation of the analyses should be strengthened, with more explicit explanation of the evidence for overfishing. Management recommendations should be clarified, citing the evidence presented. The logical step at lines 456-460 needs elaboration, and specifically how the results of the presented analyses point to the rationale for rotational management needs to be clarified.

Additional comments

The manuscript presents valuable and compelling analyses of the Cortez geoduck fishery population dynamics at a specific locale in the northern Gulf of California, and conclude that overfishing has occurred at this locale. If the authors consider the results as generalizable to the Cortez geoduck fishery, the case should be made more clearly. If they are specific to Puerto Peñasco, this information should be made explicit.
The title could be more accurate – e.g. the use of the length based stock assessment approach reveals overfishing in the species.
Introduction
Readability would be improved by reworking the paragraph at lines 94-111 as the first paragraph (more general info on stock assessments, then issues with sedentary/aggregator species as Orensanz et al outlined). The introduction should be shorter—some of the information appears extraneous to the subject of the paper (e.g. lines 123-130, though this information could be relevant to Discussion), and some would fit better in the Discussion (e.g.
Lines 88-89—this is a good discussion point). It would serve the reader to include changes in management that have already occurred, e.g. the two species in Mexico were previously managed as one species (Aragon-Noriega et al 2012).

Subheadings in Methods and Results should be mirrored and in the same order, as appropriate, and should also be reflected in the abstract (if included there). For example, the abstract lists, in order: population changes, individual growth patterns, recruitment, selectivity, fishing mortality, and survival over time. Methods lists population dynamics, fishing mortality and selectivity, recruitment, parameter estimation, and management quantities. Results lists shell length structure, recruitment, total and vulnerable biomass, harvest rate, separable fishing mortality, and finally, selectivity. It would greatly benefit the reader to be consistent with the organization of the manuscript.

Discussion
The first paragraph should clearly state the main finding of the work—present and interpret the evidence for overfishing of this population. The first three paragraphs in the Discussion are about recruitment and recruitment failures, but the focus of the paper (based on the title, abstract, and introduction) is to show new evidence, from the length-based stock assessment approach, of overfishing. The fourth paragraph of the Discussion should be expanded to include more discussion of fishing selectivity, and include criticisms of the 130 mm minimum size (Aragon-Noriega 2015; also the issue of fishery discards). Did the behavior of fishers with regard to sublegal size geoduck change over time, i.e. were they more compliant during 2010-2012 than in 2014-2016? Were sublegal size geoduck discarded on the bottom and not reported? A comparison with other fisheries that are likewise vulnerable to serial depletion would be warranted here. Management implications of the size limit, especially given that in some areas the modeled asymptotic size is below 130 mm, would also be valuable to include.

The Discussion paragraph on fishery management should closely follow the stated purpose of the study “is to analyze the effects of fishing pressure on the population of the geoduck clam, Panopea globosa, in the upper Gulf of California.” While the shortcomings of the current management strategy are outlined (although it should also be noted that P. globosa and P. generosa were managed as a single species, Aragon-Noriega et al. 2012), the interpretation of specific information provided by the current study that sheds light on management considerations should be made explicit--the sentence, “the ICSA model used in this study allowed us to estimate the changes in biomass, recruitment, and harvest rate along the shell length frequency distribution by year.” needs to be expanded to explicitly describe the interpretations that these estimations provide.

Discussion lines 441-458 are important, and should be more clearly devoted to the effect on the analyses of the new beds/areas that were exploited in 2016. Figure 9 is key to understanding some of the results. The reader needs to better understand how the results that are interpreted as indications of overfishing may be confounded by the vagaries of serial depletion of sedentary, aggregating species. Can the data be parsed to reveal specifically what occurred on tracts that were fished from 2011-2016? The new 2016 tracts in Figure 9, as the authors state, skew the information on overfishing when presented in aggregate.

Discussion lines 461-463—citations are needed re. low and high productivity areas.
Lines 464-466—there are extant examples of these strategies to cite for Panopea.

---

## Round 0.2 · accepted · Accept

I have carefully read the new version of your paper and I think you have taken into account all the comments suggested by both reviewers. I consider that your paper is ready to be published in Peerj. Congratulations!